# First historical genome of a crop bacterial pathogen from herbarium specimen: Insights into citrus canker emergence

Paola E. Campos[1,2], Clara Groot Crego [1], Karine Boyer[1], Myriam Gaudeul[2,3], Claudia Baider [4], Damien Richard [1], Olivier Pruvost [1], Philippe Roumagnac[5,6], Boris Szurek [5], Nathalie Becker[2☯], Lionel Gagnevin [5,6☯], Adrien Rieux [1☯]*

**1** CIRAD, UMR PVBMT, Saint-Pierre, La Réunion, France, **2** Institut de Systématique, Évolution, Biodiversité (ISYEB), Muséum national d'Histoire naturelle, CNRS, SU, EPHE, UA, Paris, France, **3** Herbier national (P), Muséum national d'Histoire naturelle, Paris, France, **4** Ministry of Agro Industry and Food Security, Mauritius Herbarium, R.E. Vaughan Building (MSIRI compound), Agricultural Services, Réduit, Mauritius, **5** PHIM Plant Health Institute, Univ Montpellier, CIRAD, INRAE, Institut Agro, IRD, Montpellier, France, **6** CIRAD, UMR PHIM, Montpellier, France

☯ These authors contributed equally to this work.
* adrien.rieux@cirad.fr

**Data Availability Statement:** The authors confirm that all data underlying the findings are fully available without restriction. HERB_1937 raw reads were deposited to the Sequence Read Archive

## Abstract

Over the past decade, ancient genomics has been used in the study of various pathogens. In this context, herbarium specimens provide a precious source of dated and preserved DNA material, enabling a better understanding of plant disease emergences and pathogen evolutionary history. We report here the first historical genome of a crop bacterial pathogen, *Xanthomonas citri* pv. *citri* (*Xci*), obtained from an infected herbarium specimen dating back to 1937. Comparing the 1937 genome within a large set of modern genomes, we reconstructed their phylogenetic relationships and estimated evolutionary parameters using Bayesian tip-calibration inferences. The arrival of *Xci* in the South West Indian Ocean islands was dated to the 19[th] century, probably linked to human migrations following slavery abolishment. We also assessed the metagenomic community of the herbarium specimen, showed its authenticity using DNA damage patterns, and investigated its genomic features including functional SNPs and gene content, with a focus on virulence factors.

## Author summary

Herbarium collections are a precious resource to plant pathologists, tracking crop diseases on specimens collected in the past centuries. In addition to indicating the presence of a disease at a specific time and locality, recent molecular technologies now allow extraction and microbial DNA sequencing from dead specimens. Despite challenges due to the degraded nature of DNA retrieved from historical samples, we were able to reconstruct the genome of a pathogenic bacterium from a 1937 herbarium specimen collected in Mauritius: *Xanthomonas citri* pv. *citri*, responsible for Asiatic citrus canker (ACC, an economically important agricultural disease controlled mostly through prophylactic and quarantine measures). Enhanced knowledge about the epidemiology and evolution of this

(SRR12792042). Consensus historical genome reconstructed for chromosome, plasmids pXAC33 and pXAC64 have also been deposited on GenBank database (CP072205-CP072207). The modern genomes used in this study have previously been published in the NCBI GenBank repository under accession numbers listed in S1 Table. Accession numbers of any previously published data used in this study are listed in Supplementary information.

**Funding:** This work was financially supported by l'Agence Nationale pour la Recherche (AR: JCJC MUSEOBACT contrat ANR-17-CE35-0009-01; https://anr.fr/), the European Regional Development Fund (AR, KB, OP, NB: ERDF contract GURDT I2016-1731-0006632; https://www.europe-en-france.gouv.fr/fr/fonds-europeens/fonds-europeen-de-developpement-regional-FEDER), Région Réunion (AR, KB, OP, NB; https://www.regionreunion.com/), the French Agropolis Foundation Labex Agro –Montpellier (AR, OP, PR, BS, NB, LG: E-SPACE project number 1504-004) & (AR, PR, BS, LG: MUSEOVIR project number 1600-004; https://www.agropolis-fondation.fr/?lang=en), the SYNTHESYS Project (LG: grant GB-TAF-6437 & AR: grant GB-TAF-7130; http://www.synthesys.info/), the COST Action (LG, BS: grant CA16107 EuroXanth supported by COST; https://www.cost.eu/) & CIRAD/AI-CRESI (AR, PR, BS, LG : grant 3/2016; https://www.cirad.fr/en/home-page). PhD of PC. was co-funded by ED 227, Muséum national d'Histoire naturelle et Sorbonne Université, French Ministry of Higher Education, Research and Innovation, France. The funders had no role in study design, data collection and analysis, decision to publish, or preparation of the manuscript.

**Competing interests:** The authors have declared that no competing interests exist.

bacterial pathogen is valuable to improve these measures. We compared the genome of this 1937 bacterial strain to a collection of modern strains, included it in a tree representing their genetic relationships, and calculated both evolutionary mutation rate and divergence times. This "forensic investigation" informs us about how and when the disease developed in the South West Indian Ocean Islands. We hypothesize that there was a single (or a few related) introduction of ACC in Mauritius in the mid-19th century, followed by expansion to the neighbouring islands.

## Introduction

Since the origins of agriculture, humanity has struggled with the incessant, devastating impact of plant diseases on food production [1]. As illustrated by the 19[th] century potato late blight epidemic caused by the oomycete *Phytophthora infestans* [2], crop pathogens have been responsible for tremendous losses, resulting in starvation for millions of human beings and massive migrations. Today, up to 40% of yield losses among major cultivated crops are associated with plant pathogens and pests with major economic impact [3]. Simultaneously, more than 800 million people remain chronically undernourished worldwide [4]. It is also widely acknowledged that the extensive use of pesticides against crop pathogens is detrimental to the environment, affects public health and threatens biodiversity [5].

In order to most effectively manage current infectious crop diseases and prevent future epidemics, a better understanding of the factors underlying pathogen emergence, adaptation and spread is necessary [1,6]. As sequencing technologies have become more accessible, genetic analyses have played an increasingly important role in infectious disease research. Whole genome sequencing of pathogens can confirm suspected cases of an infectious disease, discriminate between different strains, classify novel pathogens and reveal virulence mechanisms in a time- and cost-efficient manner [7,8]. In addition to examining individual pathogen sequences, multiple sequences can be combined within phylogenetic methods to assess population structure, elucidate evolutionary/transmission history and infer several demographic, evolutionary or epidemiological parameters [9,10]. Until recently, most studies were performed on field-sampled contemporary individuals over a time interval of maximum four decades. Although such studies grant a good understanding of the population structure and recent emergences of pathogens, such small temporal differences between samples do neither allow a thorough detection of measurable amounts of evolutionary changes, nor a reconstruction of deeper evolutionary timelines [11], leaving many questions on crop pathogen emergence unanswered. With the first studies on ancient DNA (hereafter aDNA) obtained from historical or ancient samples such as archaeological tissue remains or museum specimens [12], it became possible to explore the past from a genetic perspective.

The few studies performed on crop pathogens from herbarium specimens worldwide have emphasized the role of historical collections for understanding the evolution and epidemiology of plant pathogens [13–17]. First, the observation of disease symptoms associated with herbarium specimen information (collection date, geographic location, host species or other phenotypic traits) may allow a direct update of past disease occurrence, distribution and host range. For instance, Antonovics *et al.* [18] made use of infected *Silene* sp. historical specimens to survey the incidence of anther smut disease and showed a possible change of host range of this disease in the Eastern USA. Second, recent molecular developments have allowed a more efficient retrieval and sequencing of low quantity, short and degraded nucleic acids from historical desiccated plant tissues [19]. Historical and modern genomes can then be compared to detect changes in genetic contents and arrangement over time, such as the loss or gain of

functional genes or the change of ploidy levels, for both pathogens and their host plants [20]. Moreover, by expanding the temporal range between samples, the chance of detecting evolutionary changes, *i.e.* temporal signal, increases [21]. Compared to their most recent common ancestor, modern genomes are expected to have accumulated more mutations than their historical counterparts. These differences can be used to directly infer mutation rates, divergence time between lineages and sudden changes in genetic diversity [14,22], which can be correlated with historical and socioeconomic events. Such analyses performed on historical DNA sequences of *P. infestans* retrieved from 19th century herbarium specimens resolved the debated origin and identity of the strain that caused the 1840s late blight pandemic [14,17,23–25]. Although reconstructions of crop pathogen history using full genomic sequences have been successfully realized on historical oomycetes [14,23] and viruses [26–28], such an achievement has not yet been reported for a bacterial crop pathogen, for which only few genetic markers were previously exploited [29].

In this work, we focus on *Xanthomonas citri* pv. *citri* (*Xci*), the bacterium responsible for the Asiatic citrus canker (ACC) [29,30]. ACC causes important economic losses in most citrus-producing areas worldwide, both by decreasing fruit yield and quality, but also by limiting exportations due to its quarantine status [31]. The earliest records of ACC, dating back to 1812–1844, are in herbarium specimens from Indonesia and India [32], suggesting an Asiatic origin of *Xci* [33–35]. From there, although without direct evidence, *Xci* would have spread through multiple dispersal events over time, leading to its current broad distribution across continents and islands. In this context, a comparison of *Xci* multilocus genotypes retrieved from herbarium specimens suggested Japan as being the source of the 1911 ACC original outbreak in Florida [29]. Aiming for a refined chronology of these spreading events within the South West Indian Ocean (SWIO) area, where *Xci* diversity is well-documented [35–38], we focused on SWIO herbarium specimens.

We report the first genome of a historical bacterial pathogen retrieved from a citrus herbarium specimen collected in 1937 in Mauritius, 20 years after the first report of ACC on this island [39]. We studied the metagenomic composition of this herbarium sample and showed its authenticity as aDNA material by assessing damage patterns. Using tip-calibrated phylogenetic inferences performed with both the 1937 historical strain and a large set of modern genomes, we elucidated the emergence history of *Xci* in the SWIO islands and further analyzed its genomic characteristics, with a particular focus on virulence factors.

## Results

### Laboratory procedures & high-throughput sequencing

Herbarium specimen MAU 0015151 *Citrus* sp. from 1937, Mauritius (hereafter HERB_1937, Fig 1) was sampled from the Mauritius Herbarium collections (https://herbaria.plants.ox.ac.uk/bol/mau) and chosen for this study as the most ancient symptomatic herbarium specimen available from the SWIO area. It precedes the oldest culture available from this island by ~50 years. DNA was carefully extracted in a bleach-cleaned facility with no prior exposure to modern *Xci* DNA using an optimized protocol (see Material & Methods). Extracted DNA (yield of 0.75 ng per mg of leaf tissue) was shown with a specific and exclusive qPCR diagnostic assay [40] to contain *Xci* DNA, roughly equivalent to $3x10^5$ CFU/cm$^2$ (average $C_T$ of 32.0, $C_T$ cutoff = 35.4 and no-$C_T$ value for the negative control). Total DNA was then converted into an Illumina library, and sequencing generated 220.9 M paired-end reads with a base call accuracy of 99.90 to 99.96%. Following adaptor trimming and quality checking, insert reads were 59 ± 24 nt long and underwent four main analyses: metagenomic inference, ancient DNA authentication, comparative genomics and phylogenetic analyses, as summarized in Fig 2.

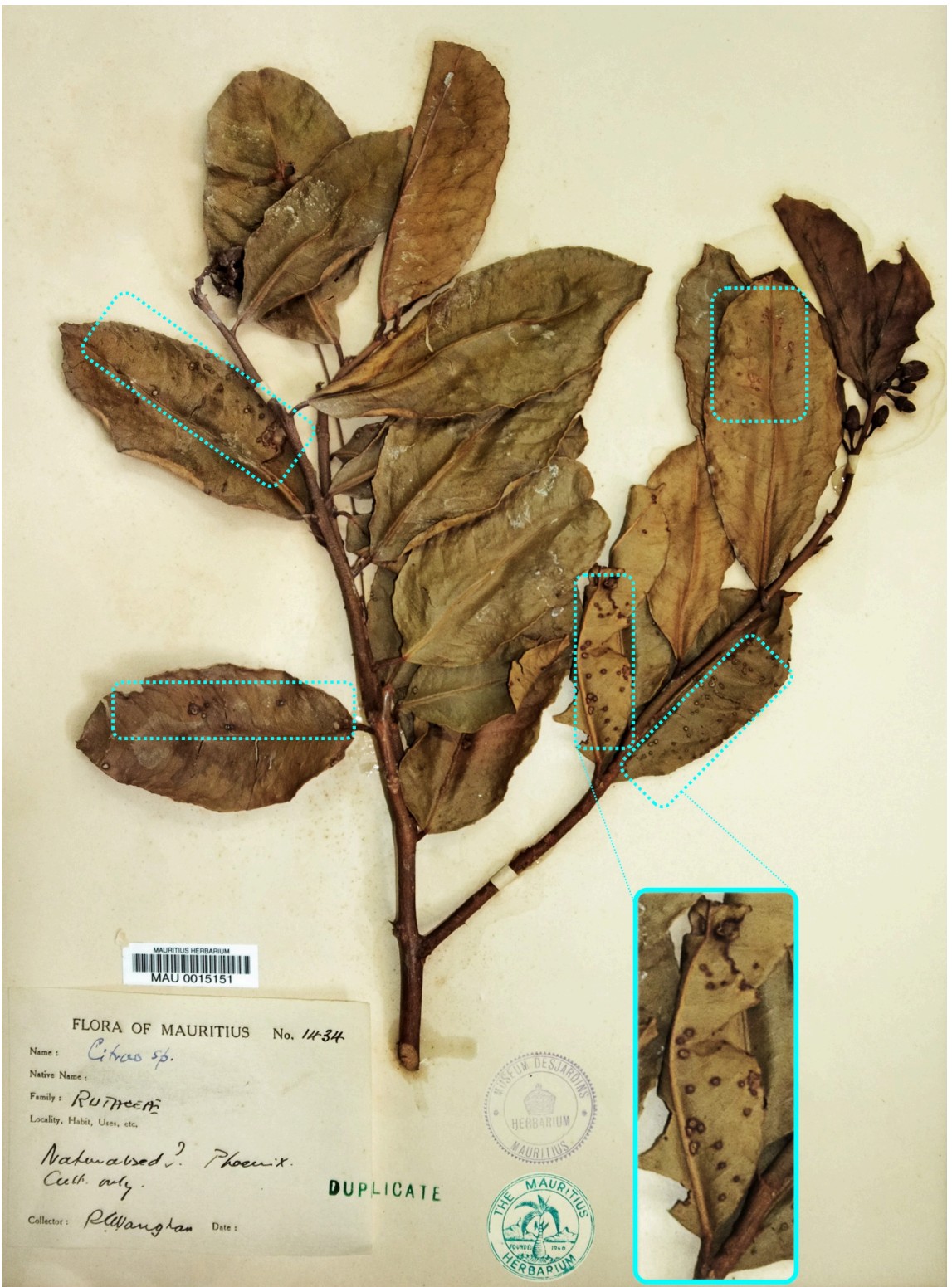

**Fig 1. *Citrus* sp. specimen MAU 0015151 (HERB_1937), Mauritius Herbarium.** MAU 0015151 *Citrus* sp. specimen (HERB_1937) was collected from Mauritius in 1937 and deposited in the Mauritius Herbarium. Leaf areas displaying typical symptoms of Asiatic citrus canker are highlighted with blue dotted frames.

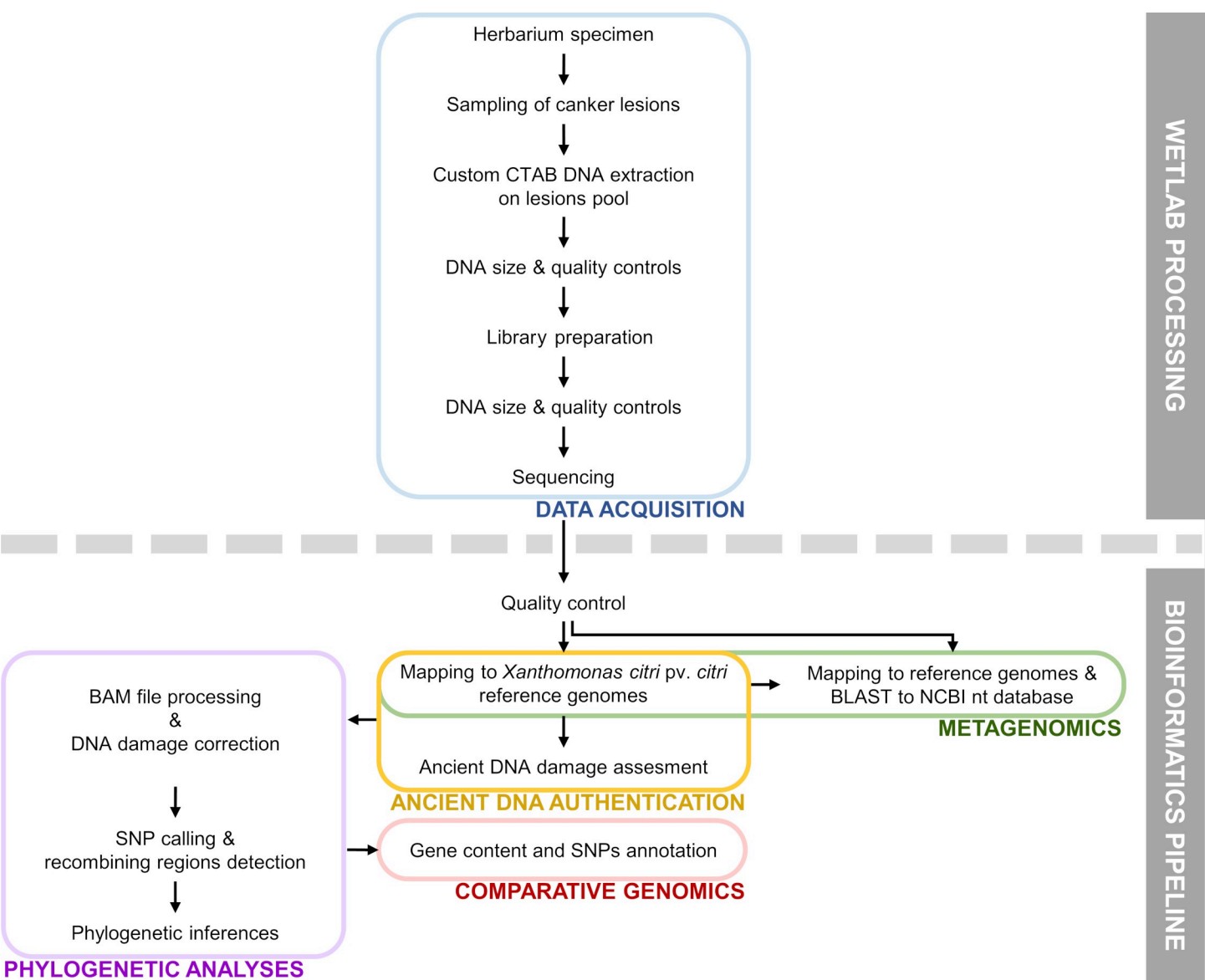

**Fig 2. Major steps performed for characterization and integration of our herbarium sample into genomic analyzes.** See Material & Methods for more details on the workflow processed for HERB_1937 in this study. Abbreviations: CTAB, cetyl trimethylammonium bromide; DNA, deoxyribonucleic acid; BAM file, binary alignment map file; BLAST, basic local alignment search tool; NCBI nt database: national center for biotechnology information nucleotide database; SNP, single-nucleotide polymorphism.

Importantly, no *Citrus* nor *Xci*-specific DNA fragments were found in our negative control, thus ruling out in-lab contamination.

## Metagenomic composition

DNA extracted from leaf lesions is expected to originate from different sources. Using a combination of mapping- and BLAST-based approaches (as detailed in Material & Methods section), we studied the metagenomic composition of the reads obtained from HERB_1937. Identified sequences mostly consisted of the *Citrus* plant host genus (21.0%), followed by, at the species level, *Homo sapiens* (5.4%), and *Xci* found in 1.2% of the reads (Fig 3). Other reads were

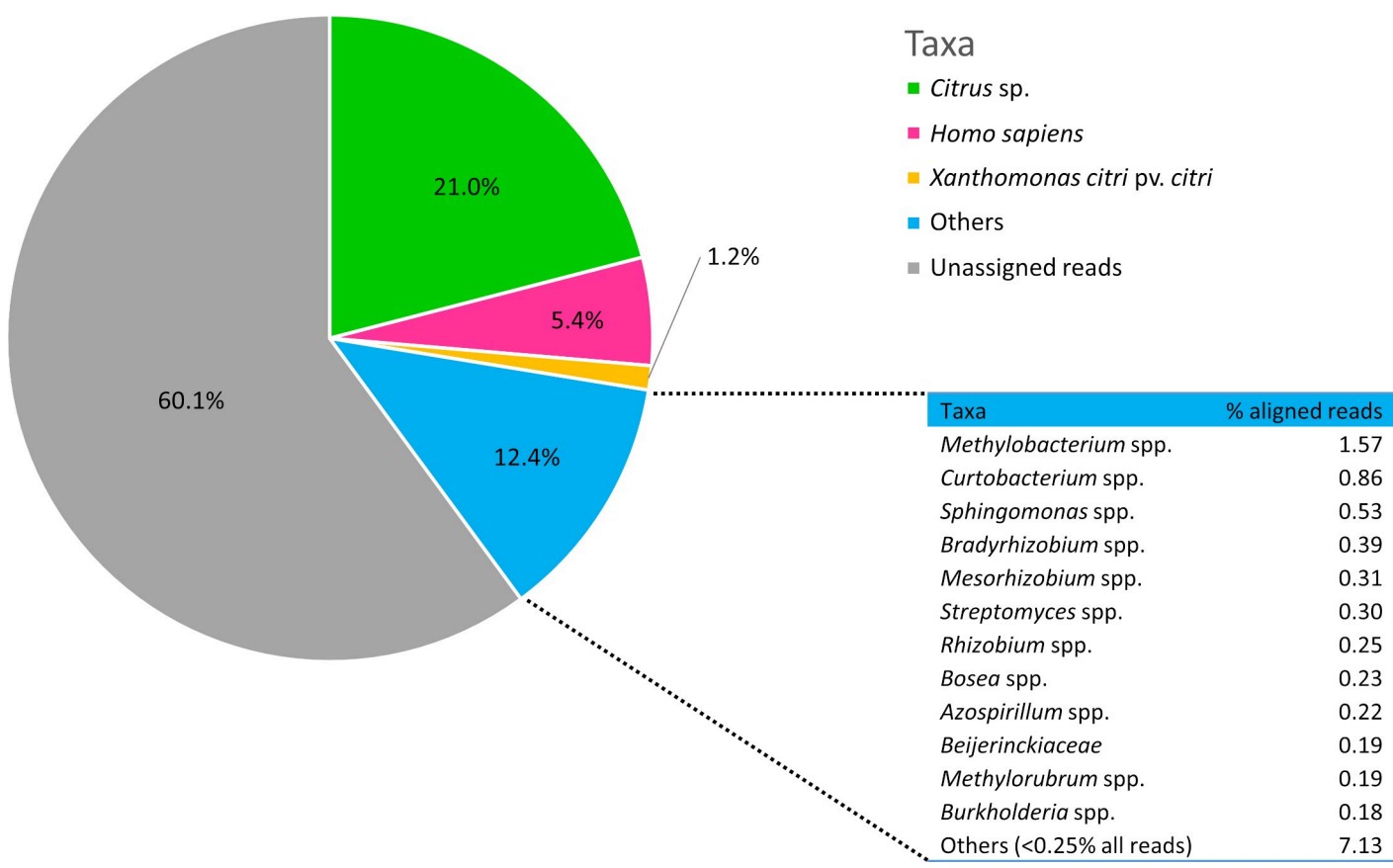

**Fig 3. Metagenomic composition of HERB_1937 historical specimen.** Proportions of reads assigned to *Homo sapiens* (5.4%), *Citrus* sp. (21.0%), *Xci* (1.2%). Others (12.4%): reads unassigned at the species level; unassigned reads (60.1%). Table: reads unassigned at the species level were assigned to the family (for *Beijerinckiaceae*) or genus level (for all others) and belong, for 0.18% to 1.57% of the aligned reads to the domain bacteria; "Others (<0.25% all reads)" include reads assigned to different plant, fungi, vertebrate, bacteria and phage genera (each identified genus totalizing less than 0.25% of all reads).

assigned to higher taxonomic levels, corresponding to one bacterial family and eleven different genera (from 0.18% (*Burkholderia*) to 1.57% (*Methylobacterium*) of aligned reads). Plant, fungi, vertebrate, bacteria and phage genera were marginally found (less than 0.25% of aligned reads for each genera). Altogether, reads unassigned to the species level added up to 12.4% of the reads. More than half (60.1%) of the reads were not assigned to any known taxa (Fig 3).

## Historical genome reconstruction & characterization

A high quality *Xci* genome was reconstructed from HERB_1937 (hereafter called HERB_1937_Xci) by mapping the reads (discarding the 5 terminal nucleotides) to *Xci* IAPAR 306 reference sequences [41]. 0.74% (N = 1,628,776) of the total number of reads mapped to the *Xci* reference genome, a value unsurprisingly smaller than the 1.2% found with the "meta-genomic pipeline" which combined both mapping and BLAST-n approaches. The reference chromosome was covered by a depth (the number of mapped reads at a given position) of at least 1X for 94% of its sequence, and displayed a mean depth of ~6X. Both pXAC33 and pXAC64 plasmids displayed a higher mean depth and larger non-covered regions (Fig 4 and Table 1). As non-covered positions can be caused by the absence of genes in the historical strain compared to the modern reference, but also by reads mapping ambiguously to multiple positions (repeated regions or replicated genes), we further characterized these loci (see gene content & virulence factors section).

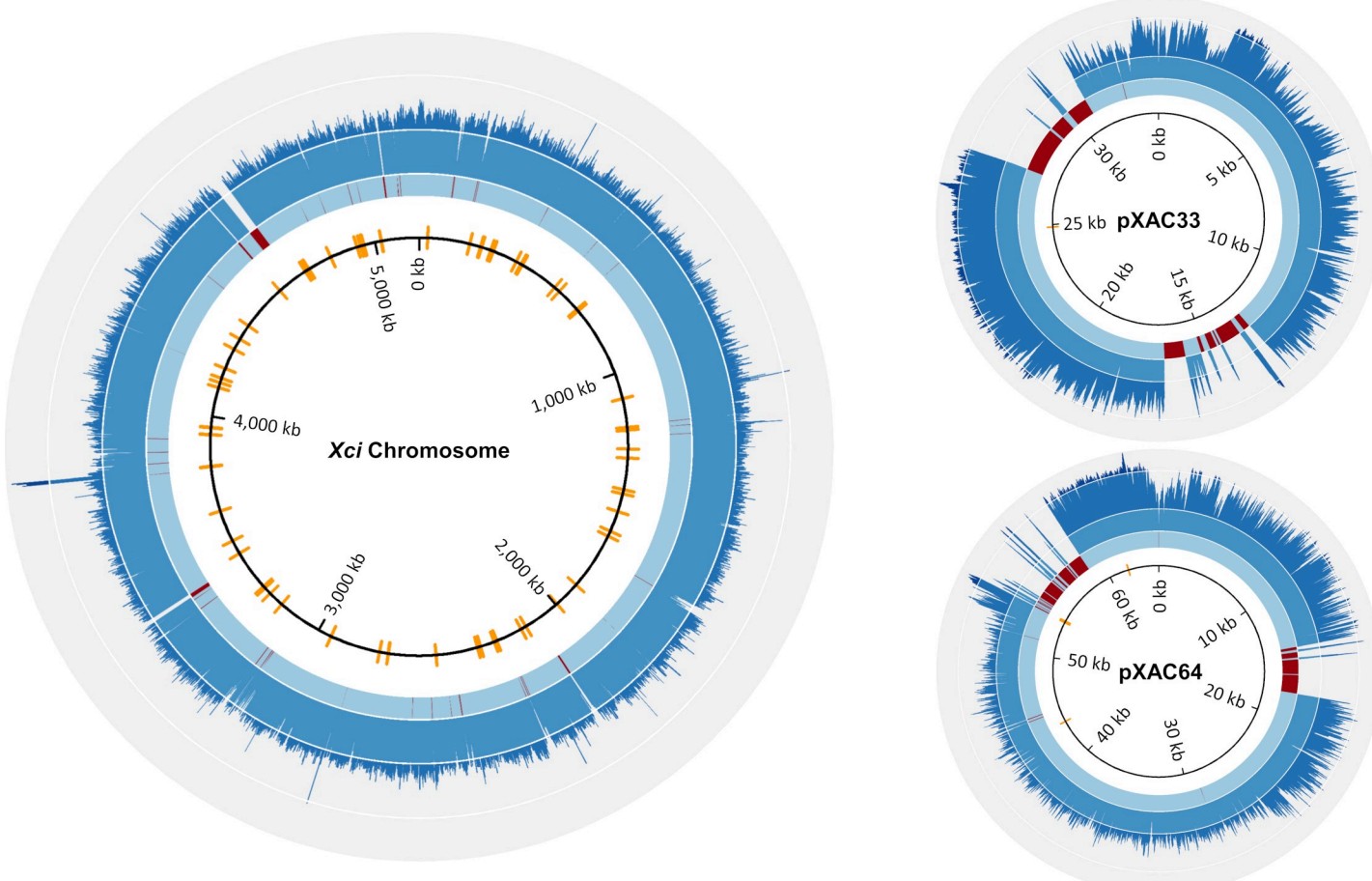

**Fig 4. Coverage plots for the reconstructed HERB_1937_Xci chromosome and plasmids (pXAC33, pXAC64) sequences.** From inside to outside, a light to dark blue scale (delimited by a white line) represents 1, 1–5, 5–15, 15-35-fold coverage (*Xci* chromosome) and 1, 1–5, 5–35, 35-90-fold coverage (plasmids). Red rings indicate no identified coverage (depth = 0). SNP positions between the respective reconstructed and reference sequences are indicated (orange line). Accession numbers for *Xci* reference strain IAPAR 306: NC_003919.1 (chromosome), NC_003921.3 (plasmid pXAC33) and NC_003922.1 (plasmid pXAC64).

**Table 1. Summary of mapping, depth coverage and damage statistics for the reconstructed HERB_1937_Xci genome.** Mapping, depth, coverage and damage statistics (read length, purine enrichment and deamination rate) are indicated for HERB_1937_Xci chromosome and plasmids (pXAC33, pXAC64). nt: nucleotides, SD: standard deviation.

| Genome | Endogenous *Xci* DNA (%)* | Mean depth** | Coverage (%)*** | | | | Read length (nt) | | Purine frequency enrichment at position -1 | | Deamination rate at terminal position (%) | |
|---|---|---|---|---|---|---|---|---|---|---|---|---|
| | | | 0X | 1X | 5X | 10X | Mean | SD | Mean | SD | 5'C/T | 3'G/A |
| Chromosome | 0.71 | 5.9 | 5.8 | 94.2 | 53.0 | 7.0 | 42.75 | 12.64 | 1.79 | 0.00 | 2.25 | 2.35 |
| pXAC33 | 0.01 | 21.9 | 17.4 | 82.6 | 80.2 | 75.1 | 45.43 | 14.21 | 1.76 | 0.05 | 2.91 | 2.96 |
| pXAC64 | 0.02 | 17.3 | 11.5 | 88.5 | 82.3 | 63.6 | 45.20 | 13.85 | 1.77 | 0.01 | 2.65 | 2.73 |
| Mean | | | | | | | 44.46 | 13.57 | 1.77 | 0.03 | 2.60 | 2.68 |

* Reads mapping to *Xci* reference genome/total reads before duplicate removal, expressed in %.

** Average number of mapped reads at each base of the reference genome.

*** Percentage of reference genome covered at nX depth.

## Ancient DNA damage assessment

Ancient DNA is typically degraded, presenting short fragments, excess of purine bases before DNA breaking points and cytosine deamination at fragment extremities [12,42]. We searched for such patterns of degradation in HERB_1937_Xci using the dedicated tool map-Damage2 [43]. The mean read length of HERB_1937_Xci reads was 44.5 ± 13.5 nt, showing substantial fragmentation (Fig 5A). DNA fragmentation being partially caused by depurination, we also examined the nucleotidic context surrounding 5' end DNA breakpoints. We found a mean relative purine enrichment of 1.77 ± 0.03 between upstream positions -1 and -5 of HERB_1937_Xci reads (Table 1). Modern strains, fragmented by enzymatic digestion prior to library construction, displayed no such enrichment (0.87 ± 0.01). Cytosine deamination was investigated by monitoring 5'C/T substitutions *versus* complementary 3'G/A substitutions, a classical analysis for double-stranded, blunt-ended libraries constructed prior to sequencing. Mean deamination rates of HERB_1937_Xci reads reached maximal values at the terminal nucleotide (2.64 ± 0.29%, Table 1 and Fig 5B). For statistical analyses, we took into account the five successive extreme positions of the reads, harbouring a significant increase between each nucleotide (outwards) along the five first or last positions of the reads (Wilcoxon matched-pairs signed rank test, 2-tailed p-value = 0.0313). The maximal rate among reads of three modern *Xci* controls displayed a significantly lower value of 0.10 ± 0.07% (p<0.0001, unpaired 2-tailed Mann-Whitney test, Fig 5B). The apparent lower deamination rate for HERB_1937_Xci chromosome reads, as compared to both plasmids (Table 1), was analyzed similarly, along the five first or last positions of the reads. Interestingly, we found a significantly lower deamination rate for HERB_1937_Xci chromosome reads (1.6%) as compared to both plasmids (1.9%) (Wilcoxon matched-pairs signed rank test, 2-tailed p-value = 0.002). In

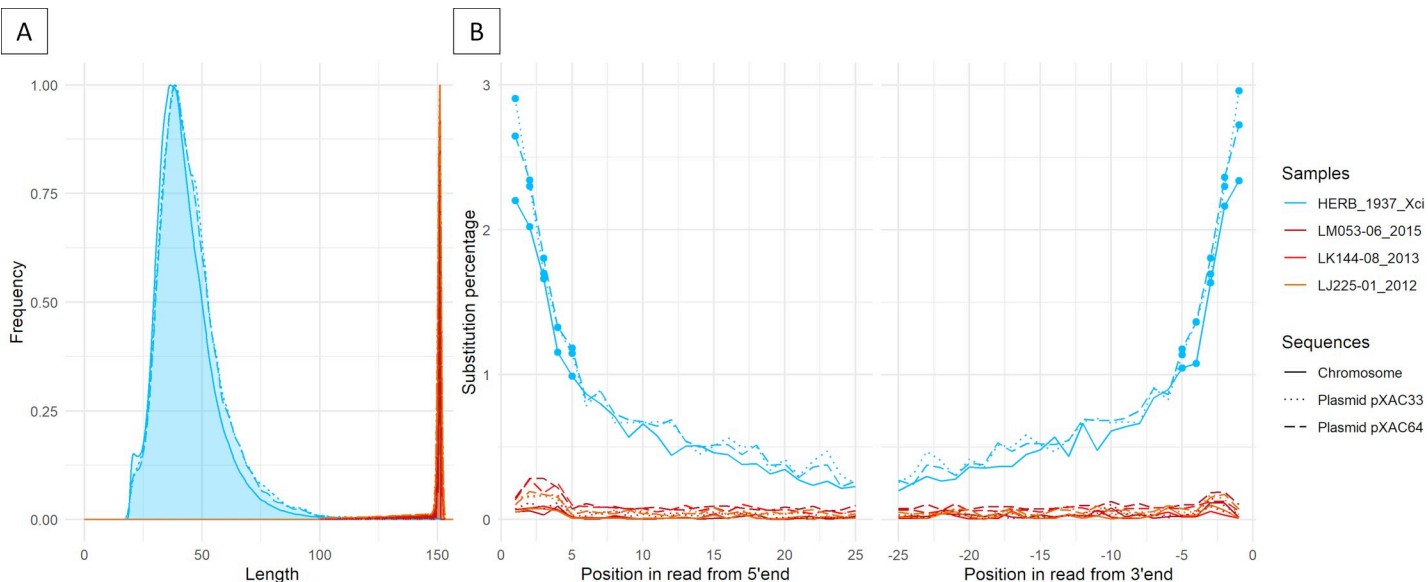

**Fig 5. HERB_1937_Xci post-mortem DNA damage patterns.** Post-mortem DNA damage patterns were measured on historical HERB_1937_Xci (full, dotted or dashed blue lines for chromosome, pXAC33 and pXAC64 respectively) and compared with three modern *Xci* strains isolated from SWIO in 2012, 2013 and 2015 respectively (red lines, see results and S1 Table for full description). (A) Fragment length distribution (nucleotides; relative frequency in arbitrary units). (B) Deamination percentages of the first 25 nucleotides from the 5' (C to T substitutions) and 3' (G to A substitutions) ends, respectively. Dots: five most extreme nucleotides of the reads, showing a significant increase (towards the extremity) between each nucleotide along the five first or last positions of all HERB_1937_Xci reads. Along the five extreme nucleotides, reads matching to HERB_1937_Xci harboured significantly higher values than modern controls, and reads matching to the HERB_1937_Xci chromosome harboured significantly lower deamination rates than sequences matching to either plasmid (see results for statistics).

contrast, we observed similar fragment lengths for chromosome and plasmid reads (Table 1 and Fig 5).

We performed the same analyses using the *Methylobacterium* reference genome (*M. organophilum*, strain DSM 760), since 1.57% of the reads unassigned at the species level were attributed to this particular genus (Fig 3). Mean read length was estimated at 66 ± 25 nt, relative enrichment of purine frequency reached 1.80 ± 0.52 (as previously, between position -1 and -5), and deamination rates at the terminal nucleotides averaged 2.18 ± 0.07%.

## Gene content & virulence factors

Out of 5,125 coding sequences (CDS) of strain IAPAR 306, only 139 were covered on less than 75% of their length by HERB_1937_Xci reads and will hereafter be designed as non-covered (S2 Table). Ninety-five of those CDS are present in multiple copies in IAPAR 306 with a strong nucleotide identity, leading to ambiguous mapping and poor coverage and can be considered as false absences. Among those 95 CDS, 77 are predicted to encode for full-length, or fragments of transposases. Four correspond to highly identical copies of Transcription Activator-Like Effector (TALE) genes (see specific paragraph below). The remaining multi-copy genes code for the elongation factor Tu, a xylose isomerase, a filamentous haemagglutinin and seven hypothetical proteins. Forty-four IAPAR 306 CDS were non-covered because no homologous reads were present in our dataset. Most of them hypothetically code for proteins of unknown function, with the exception of an identified type I restriction-modification system (including DNA methylase, endonuclease and specificity determinant), and six recombinases or integrases. Interestingly, 28 successive non-covered CDS correspond to a 27-kb block present only in IAPAR 306 and a few of its close relatives, which contains six transposases, three recombinases and 19 proteins of unknown function.

We verified the presence or absence of specific genes whose products have proven or are hypothetically involved in the pathogenicity of *Xanthomonas*. In particular, the type III secretion system (T3SS) is a syringe-like apparatus which injects "effectors" directly into the plant cell to inhibit plant defences and contribute to symptom development [44]. For this we investigated the presence in HERB_1937_Xci of a group of 82 genes found in *Xci* or other *Xanthomonas* species [45], encoding either for the T3SS or for type III effectors (T3E, which may participate in pathogenicity) [46]. Reads from HERB_1937_Xci covered 57 of those CDS on more than 94% of their sequence (S3 Table), including the entire set (24 CDS) of genes necessary for the T3SS and 33 potential T3E genes. The coverage of the remaining 25 CDS, all virulence factors from other *Xanthomonas* but not present in *Xci* [46] reached a maximum of 45.1% of their length (S3 Table), indicating the absence of the corresponding genes.

On the plasmids, most of the non-covered positions of HERB_1937_Xci were localized in four regions coding for TALE proteins (Fig 4 and S2 Table). These peculiar T3Es are responsible for the development of canker symptoms on citrus [47]; as our samples harbored such symptoms, we expected to find homologs in HERB_1937_Xci. *Xci* injects these TALE into the plant cell, activating the host's transcriptional machinery to its benefit [48,49]. *Tale* genes encode for transcription activator-like proteins containing an N-terminal domain responsible for translocation from the bacterium to the plant cell, a C-terminal domain containing nuclear localization signals and a eukaryotic transcription activation domain, flanking tandem repeats of 33–34 conserved amino acids (S1 Fig). These repeats are highly homologous except for two amino-acid residues (called Repeat Variable Di-residues: RVD) responsible for DNA-binding specificity. We hypothesized that reads corresponding to *tale* genes were initially not mapped due to their particular structure and multicopy nature. Hence, we realized specific alignments using the sequences coding for either the N-terminal domain, the C-terminal domain and the

repeat domain (reduced to a three-repeat string) of *tale* gene *pthA4* as three independent references to test for the presence of *tale* sequences in HERB_1937_Xci. Almost 6,000 newly mapped reads corresponding to the *tale* gene were recovered (S1 Fig), with a mean depth of 44X for both 5' and 3' ends, about two times the mean depth of plasmid sequences outside *tale* gene positions (~23X). Moreover, two loci on the 5' end sequence were biallelic, presenting either T or C bases with a T/C ratio of 43/57 and 35/65, respectively. One of these loci translated into a conservative amino-acid substitution, found elsewhere in TALEs of proteic databases. Taken together, these results suggest the existence of two to four different 5' end sequences of *tale* genes, and therefore as many *tale* genes, in HERB_1937_Xci historical genome. Finally, among the remaining reads corresponding to the central repeat domain, we identified eight patterns of nucleotides coding for the RVD (S4 Table). Interestingly, although the most prevalent are found in modern *Xci tale* genes in similar proportions [50], three RVDs are unreported in modern TALE.

## SNPs, phylogenetic reconstruction and tip-dating at the SWIO scale

We localized the SNPs between HERB_1937_Xci and the IAPAR 306 reference genome (Fig 4). After filtration of dubious SNPs (i.e. eliminated because of low depth, heteroplasy and/or proximity to another SNP), HERB_1937_Xci displayed 83 high-quality SNPs on its chromosome sequence, one and four in sequences corresponding to pXAC33 and pXAC64, respectively. Forty-three SNPs were non-synonymous substitutions on the chromosome reference sequence, one on pXAC33 and three on pXAC64. The SNPs found between HERB_1937_Xci and IAPAR 306 were not characterized any further; a more meaningful analysis was performed for SNPs identified between HERB_1937_Xci and its related SWIO clade.

A total of 2,634 high confidence SNPs were found within the alignment of HERB_1937_Xci historical chromosome with 116 modern samples from the SWIO islands. The ClonalFrameML [51] analysis identified a single 5.9 kb recombinant region including two SNPs, which were removed from further inferences. For the 2,632 recombination-free SNPs, we estimated a ratio of non-synonymous (d*N*) to synonymous (d*S*) changes of 4.16. We identified 15 SNPs unique to HERB_1937_Xci and restricted to chromosome sequences, among which 14 were attributed to coding sequences. Analysis of these SNPs led to the identification of three synonymous and 11 non-synonymous mutations, which were characterized at the protein level. Interestingly, seven of those reveal unique amino-acids at these positions (among similar but non-redundant proteins of the *Xanthomonas* genus identified by BLASTp), thus harboring previously unknown proteic features (S5 Table). We added an outgroup and built a Maximum-Likelihood (ML) phylogeny with RAxML [52], which placed HERB_1937_Xci historical sequence outside of the "modern" SWIO clade (S2 Fig). The ML tree was well-supported and structured in three lineages: a Mauritius lineage (lineage A), sister-group of the rest of the modern strains of the SWIO comprising two lineages, the first with strains from Mauritius and Reunion (lineage B), and the second with strains from all SWIO islands (lineage C) (S2 Fig).

As a requirement to perform tip-based calibration, we tested the presence of temporal signal in our tree with both a linear regression between samples ages and root-to-tip distance, and a date-randomization test [22]. Both statistical tests revealed the presence of temporal signal (*i.e.* progressive accumulation of substitutions over time) within the SWIO tree. The linear regression test displayed a significant positive slope (value = $19.236 \times 10^{-5}$, adjusted $R^2 = 0.270$ with a p-value = $2.07 \times 10^{-10}$), with HERB_1937_Xci showing clear evidence of branch shortening (Fig 6B). The date-randomization test of the inferred root age of the real *versus* date-randomized datasets showed no overlap (95% Highest Posterior Density, S3 Fig). Therefore, we

built a time-calibrated tree with BEAST [53], which was globally congruent (similar topology and node supports) with the ML tree (Fig 6A). Phylogenetic diversity of Mauritius island strains (1530.4) was significantly higher (p-value = $2.2 \times 10^{-16}$) than those calculated from the other islands (Reunion strains = 1518.7, Rodrigues = 691.1, Comoros = 1024.0 and Mayotte = 319.0). We inferred a root date of 1843 [95% HPD: 1803–1881] and a mean substitution rate of $9.4 \times 10^{-8}$ [95% HPD: $7.3 \times 10^{-8}$–$11.4 \times 10^{-8}$] per site per year, with a standard deviation for the uncorrelated log-normal relaxed clock of 0.271 [95% HPD: 0.182–0.366] suggesting low heterogeneity amongst branches (Figs 6B and S4). To specifically evaluate the contribution of HERB_1937_Xci, we considered modern strains only: although the dataset still displayed temporal signal (slope value = $9.885 \times 10^{-5}$, adjusted $R^2$ = 0.077, p-value = 0.0009) (Fig 6B), the BEAST analysis performed under the same parameters yielded significantly different values. An older tree root date of 1800 [95% HPD: 1745–1852] was inferred, together with a lower mean substitution rate of $8.2 \times 10^{-8}$ substitutions per site per year [95% HPD: $6.4 \times 10^{-8}$–$9.9 \times 10^{-8}$] and a standard deviation for the uncorrelated log-normal relaxed clock of 0.188 [95% HPD: 0.082–0.289] among branches (S4B Fig). In summary, when comparing the estimates of root ages–with and without HERB_1937_Xci in the datasets–our results indicate that integrating the historical sequence significantly improves the accuracy of the temporal inferences, with a reduction of the 95% HPD from ~107 years to ~78 years (Fig 6C).

## Discussion

We sequenced the genome of HERB_1937_Xci, an historical strain of the crop bacterial pathogen *Xanthomonas citri* pv. c*itri* (*Xci*) from an infected herbarium specimen sampled in 1937 in Mauritius. To our knowledge, HERB_1937_Xci is the first historical genome of a pathogenic bacterium obtained from herbarium material. Similar achievement has been previously successfully realized on viruses [28,54], oomycetes such as *Phytophthora infestans* [14,23,24], and more recently on cyanobacteria [55]. But for plant pathogenic bacteria in general, and more specifically for *Xci*, only multilocus genotyping data could be exploited from such historical material [29].

Adopting a shotgun-based deep sequencing strategy allowed us to describe the metagenomic diversity contained within our historical herbarium specimen. Among assigned reads, HERB_1937 displayed 1.2% of *Xci* DNA for 21.0% of *Citrus* sp. DNA, a pathogen/plant ratio in the range of those previously observed for *P. infestans*, a nonvascular pathogen isolated from infected herbarium potato leaves [20,23,24]. The microbial community also contained several bacterial genera, all described in NGS studies as part of the citrus leaf [56] or root [56,57] microbiota. The three most prominent genera (*Methylobacterium*, *Curtobacterium* and *Sphingomonas*, >0.5% of aligned reads) belong to the core citrus leaf microbiome [58], and the relative abundance of *Methylobacterium* reads among bacteria (11.7%) is consistent with studies on modern samples (from 5 to 58% [56]). These bacterial genera were thus likely associated to the living citrus plant, and/or to HERB_1937 sample, colonized during collection and storage in the herbarium. Bieker *et al.* [59], using deamination studies, identified a fungal species proposed to have colonized herbarium specimens shortly after collection. As illustrated by the typical aDNA patterns we observed for *Methylobacterium* spp., we may exclude recent or laboratory contaminations [60]. Interestingly, up to 5.4% of the reads were assigned to human DNA resulting from contaminations during specimen manipulation (collection, mounting or storage). Finally, a substantial amount (60.1%) of HERB_1937 were unassigned reads, reflecting either the incompleteness of the reference database as compared to the microbial diversity of the sample [61] or the difficulty for short reads to be assigned taxonomically, a typical result in ancient DNA research [62].

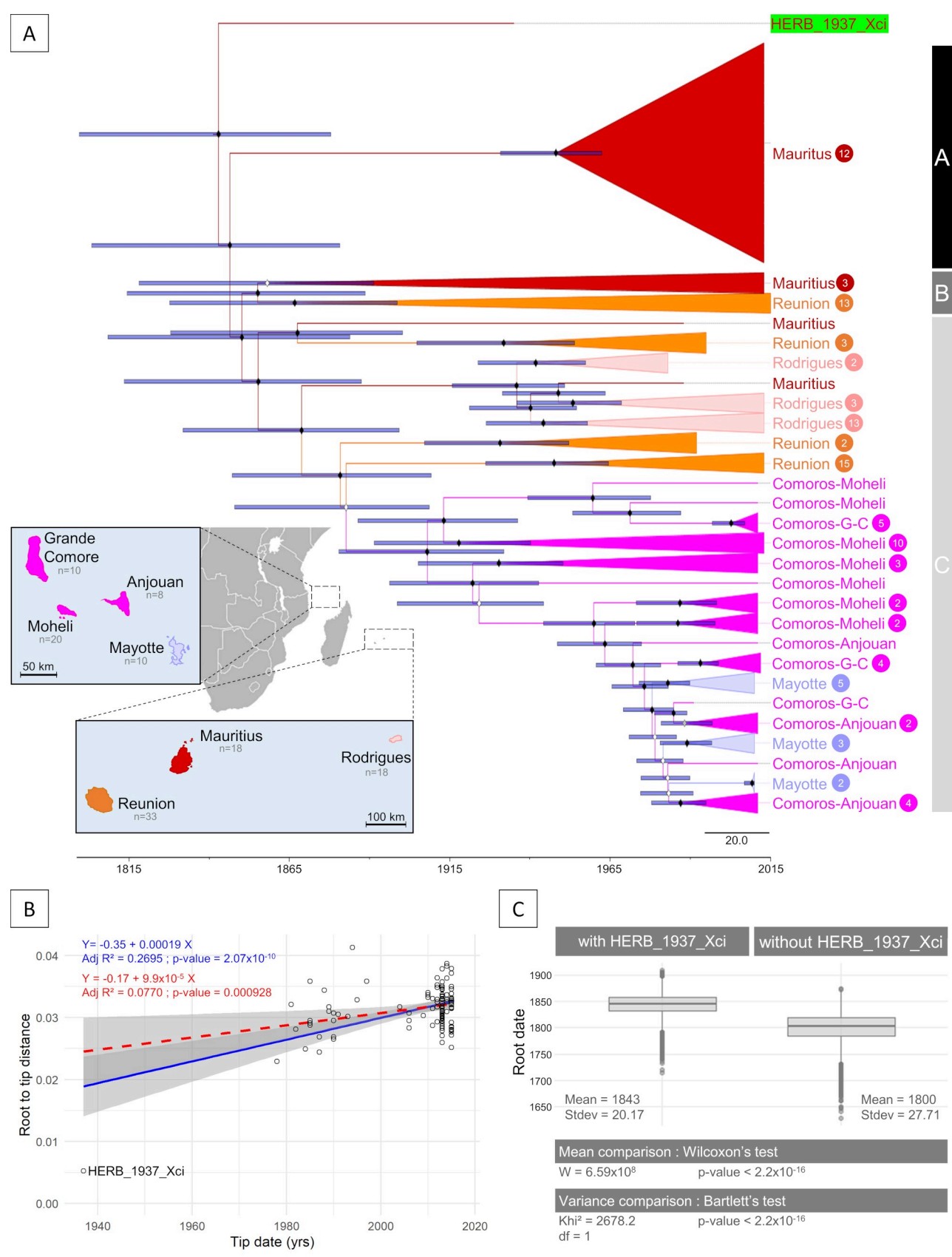

**Fig 6. Tip-dating Bayesian inferences on historical and modern *Xci* genomes from the SWIO islands.** (A) Dated BEAST tree of 116 *Xci* modern strains sampled from the SWIO islands between 1978 and 2015 with historical HERB_1937_Xci (highlighted in green) built from 2,632 non recombining SNPs. Node support values are displayed by diamonds, in white for Posterior Probabilities below 0.9, in black for values above 0.9; node bars cover 95% Highest Probability Density of node height. The tree is structured in three lineages (A, B & C). Branches are collapsed and colored, according to the sample's geographic origin, except lineage A, which is cartooned to help visualization. Tip labels include the geographic origin and, in cases of collapsed or cartooned branches, the number of samples. Map layer is from Natural Earth, available from https://www.naturalearthdata.com. (B) Linear regression of root-to-tip distance on year of sampling (tip date) test for temporal signal. Regression lines are plotted in blue when integrating historical HERB_1937_Xci genome and in red (dotted lines) when not. Grey areas correspond to their confidence interval. Associated values are the regression equation, adjusted $R^2$ (Adj $R^2$) and p-value. (C) Boxplot distribution of root age, with (left) and without (right) integrating historical HERB_1937_Xci in the dating inference, and associated statistical comparisons. Boxes represent 25th to 75th percentiles, Minimum-Maximum intervals are displayed by a vertical bar and outliers as circles.

Characterization of DNA degradation patterns specific to aDNA (fragmentation, depurination and deamination) combined with clear evidence of branch shortening confirmed the historical nature of our reconstructed *Xci* genome, a key point in any ancient DNA study [12,41]. Patterns of DNA degradation of HERB_1937_Xci appeared consistent with those measured on *P. infestans* from 19th century herbarium samples [23,24,61]. Interestingly, we observed significantly higher deamination rates of cytosine residues in reads mapping to either of the two plasmids, as compared to reads mapping to chromosomal DNA. Depurination rates and fragment sizes did not harbour such significant differences in our study. A possible explanation for our observation relates to differential methylation patterns of cytosines. In a recent study investigating epigenetic modifications in *Xanthomonas* species, N4-methylcytosines (N4meC, a bacteria-specific pattern) were identified in higher proportions in chromosomes *versus* plasmids [63]. Interestingly, N4meC have previously been found to be more resistant to deamination than unmethylated cytosines [64]. The lower deamination rate observed on HERB_1937_Xci chromosome (as compared to the plasmids) could thus be due to a better protection of the chromosomal cytosines from deamination, independently of depurination and fragmentation mechanisms. Further investigations, such as ancient methylome mappings [65], should refine our molecular understanding of the degradation patterns observed in this study.

Phylogenetic reconstruction confidently placed HERB_1937_Xci at the root of the modern SWIO lineages, a position that reveals its genetic relatedness with the SWIO *Xci* founding population. Modern Mauritian strains were found in all three main SWIO lineages and displayed the highest phylogenetic diversity, a typical pattern for source populations during biological invasions [66], which points Mauritius as the most likely entry point of ACC disease in the SWIO islands. Future studies including new historical genomes from other islands will be necessary to confirm this hypothesis. We estimated the age of the ancestor of all strains (*i.e.* the root), which is a proxy for *Xci* emergence date to 1843 [95% HPD: 1803–1881]. This predates the earliest record of the disease in the area (1917 in Mauritius [39]) and refines the recent estimation of 1818 [95% HPD: 1762–1868] obtained from modern strains only [36]. *Xci* and its main host genus, *Citrus*, originated in Asia [34,35] and were most likely disseminated out of their area of origin by human-mediated movements of plants or plant propagative material [31,67]. Richard *et al.* proposed two possible origins of the pathogen in the SWIO [36]. On the one hand, they hypothesized that a French botanist and colonial administrator, Pierre Poivre (1719–1786) could have introduced infected citrus plants from several Asian countries during his numerous peregrinations starting in mid-18th century [68]. Later, tens of thousands of indentured labourers arrived from several Asian countries (most numerously from India) after the abolition of slavery in Mauritius (1835) and Reunion (1848), mainly to work in agriculture [69]. This flow of people from the Asiatic continent, along with their possessions which consisted among other things of seeds, plants and fruits [70] may have led to the introduction of *Xci* in the SWIO area. The updated time frame of emergence inferred from our data favours

the second scenario. Future work including strains from the hypothetical Asian cradle of *Xci*, with some possibly obtained from herbarium specimens, will be required to investigate the geographic origin of the strains that first invaded SWIO.

Although both the root position of HERB_1937_Xci and the monophyly of all SWIO strains suggest one or few successful historical introductions of genetically (and likely geographically) closely related *Xci* strains in this area, the structure of the phylogeny indicates multiple inter-island migration events, likely via infected plant material exchange. Such events may have first occurred between Mauritius and Reunion islands at the very beginning of the history of *Xci* in the area, as illustrated by the deepness of the most recent common ancestor (MRCA) shared between strains of those two islands that used to share tight historical and political links at the time. More recent migrations between *i*) Mauritius and Rodrigues (an island ca. 600 km east of Mauritius, part of its territory), *ii*) Reunion and the Comoros archipelago (Mayotte, 1,435 km distant from Reunion, part of the French overseas territories) and *iii*) the four islands of the Comoros archipelago (promoted by their historical and economic relationships, see insert Fig 6A). Altogether, our findings emphasise the influence of human-associated migratory events in shaping the global distribution and the emergence of preadapted crop pathogens, a well-known phenomenon [6,24,71,72]. Additionally, our results indicate that integrating historical genomes in phylogenetic analyses significantly refines divergence time estimates, as highlighted in previous ancient DNA studies [73,74].

Tip-date calibration of the SWIO phylogenetic tree also enabled us to estimate a mean mutation rate of $9.4 \times 10^{-8}$ substitutions per site per year for *Xci*. This value is consistent with the recent estimation of $8.4 \times 10^{-8}$ substitutions per site per year obtained by Richard *et al.* in the same area [36] and falls within estimations made over a similar time span (80 years) on several human-associated bacterial pathogens, spanning one order of magnitude ($10^{-8}$–$10^{-7}$) [75]. This rate, among the first published for a crop pathogen, is averaged across all sites of the non-recombining portions of the *Xci* chromosome and appears to be homogeneous within the various SWIO lineages. Interestingly, we observed a relatively high d$N$/d$S$ ratio as compared to other bacterial species [76], which might result from selection for diversification following *Xci* emergence and evolution within SWIO islands. In summary, our substitution rate estimate is crucial to further studies, since it can improve the prediction of the evolution of *Xci* using various modelling-based frameworks.

Finally, we aimed to compare the genomic features of HERB_1937_Xci with its modern counterparts. Among the 15 SNPs unique to HERB_1937_Xci and restricted to chromosome sequences, five non-synonymous SNPs are considered to induce conservative amino acid changes, and are thus not expected to alter the conformation or the active site of their respective proteins. Interestingly, among the six non-conservative SNPs, the location (next to the hinge and binding domain) of an amino acid substitution of the essential metabolic enzyme isocitrate dehydrogenase could modulate its adaptability, and thus the fitness of the pathogen [77]. Finally, seven non-synonymous SNPs account for unique amino acids at given positions of *Xanthomonas* sequence alignments, providing an exclusive signature for seven HERB_1937_Xci specific protein homologs.

Our investigation of HERB_1937_Xci gene content showed that it was globally similar to the one observed in reference strain IAPAR 306. The non-covered CDS corresponded to repeated CDS or to absent CDS. The former are mostly transposases, or other multicopy genes. Among actually absent CDS are mostly proteins of unknown function, recombinases, or notably a type I restriction-modification system, together with four adjacent CDS. The 27-kb block probably corresponds to a genomic island recently acquired by strain IAPAR 306 (and a few of its close relatives) but absent in other *Xci*. It is inserted in the middle of a CDS encoding for a putative competence protein [78]. However, as our gene content analysis

resulted from sequences reconstructed by mapping, we were unable to identify potential genome rearrangements. Furthermore, any genetic material present in HERB_1937_Xci but absent in the reference sequences used to reconstruct the historical genome would have been missed. Gene content investigation based on *de novo* assembly of historical reads would be a way to overcome this limitation but the short length of aDNA reads, their mixed origin as well as the relatively low coverage of HERB_1937_Xci hampered us from applying such strategy [79,80].

We showed that HERB_1937_Xci contains a complete set of genes for its type III secretion system, as well as the same assortment of effector genes as modern *Xci* strains [46]. In particular, Transcription Activator-Like Effectors (TALE) are crucial virulence factors for *Xci* [50]. We determined HERB_1937_Xci to possess between two and four paralogs of the functional *tale* gene *pthA4* present in strain IAPAR 306, a value consistent with modern *Xci* strains [81]. Although it was not possible to localize them in the genome or reconstitute their central repeat domain, sequences corresponding to their N- and C-terminal domains, as well as a repertoire of RVD sequences, were reconstructed, suggesting their functionality. Most of the essential RVD sequences were present in approximately the same proportion as in modern *Xci tale* genes. Three unique sequences encoding unreported RVDs could be mutational variants of the present RVD sequences: AAA—from AAT—(K* from N*), CACGAA and CACGGAT from CACGAT (respectively HE and QD from HD). A design of TALENs with artificial RVDs recently showed that HE and QD were functional and preferentially binding to C on target DNA *in vivo*, like HD [82]. This suggests that apart from undetectable loss of genes (in the case of effectors not identified in databases) and modification in the structure of the repeat region of TALE genes (which might have an important impact on virulence), the effector repertoire of *Xci* has been stable at least since the time of the last common ancestor of all SWIO strains.

In summary, our results show that herbarium specimens can provide a wealth of genomic information on bacterial pathogens, their associated microbial community, or their plant host (an aspect that we did not explore in this study). The present work focused on a single herbarium specimen in order to evaluate the feasibility of genetic analyzes and the added value such samples bring to phylogenetic and epidemiological approaches. Broader studies to reconstruct Asiatic citrus canker's worldwide propagation and evolutionary history would require additional, well-chosen, geographically and temporally representative samples. More generally, similar investigations could be and are performed on other important bacterial plant pathogens to elucidate their evolutionary history, investigate plant-pathogens interactions further, and study the temporal dynamic of plant-associated microbial communities. Such studies emphasise the interest of biological collections and will hopefully help to decipher the epidemiological and evolutionary factors leading to the emergence of plant pathogens. This, in turn, may provide clues to improve disease monitoring and achieve sustainable control.

## Material & methods

### Herbarium sampling

The collections of the Mauritius Herbarium (https://herbaria.plants.ox.ac.uk/bol/mau) were prospected in June 2017. Several citrus specimens displaying typical citrus canker lesions were sampled on site using gloves and sterile equipment and brought back to CIRAD laboratory in individual envelopes where they have been stored in vacuum-sealed boxes at 17°C until use. MAU 0015151 (Fig 1), a *Citrus* sp. specimen collected by Reginald E. Vaughan at Phoenix, Mauritius in 1937 was chosen as being the oldest specimen sampled from the SWIO area. The date and exact place of collection, which do not appear on the specimen itself, were found in the original collection book of the collector. MAU 0015151 was deposited in 1937 at the

collection of the Mauritius Institute (Port Louis, Mauritius). This collection was moved in 1960 to Réduit to form the core of The Mauritius Herbarium (MAU, acronym according Thiers 2021), where it has been preserved since under controlled temperature and humidity and regularly poisoned (e.g. fumigation and/or use of Kew mixture: solution of mercury, phenol, and ethanol).

## DNA extraction, quality control and real-time quantitative PCR assay

HERB_1937 sample DNA extraction was performed in a bleach-cleaned facility room with no prior exposure to modern *Xci* DNA. DNA extraction was performed following a custom CTAB protocol modified from Ausubel (2003) [83]. Briefly, a pool of five canker lesions (to obtain approximately 10 mg) from a single leaf of HERB_1937 were cut. A 10 mg piece of a plant species that is not a host to *Xci*, a *Coffea arabica* herbarium 1965 specimen, was integrated as an aDNA negative control sample. Both samples were pulverised at room temperature and soaked in a CTAB extraction solution (1% CTAB, 700 mM NaCl, 0.1 mg/mL Proteinase K, 0.05 mg/mL RNAse A, 0.5% N- lauroylsarcosine, 1X Tris-EDTA) under constant agitation and until tissue lysis at 56˚C (up to six hours); an equal volume of 24:1 chloroform: isoamyl alcohol was added before centrifugation and recuperation of the aqueous phase (twice), followed by adding 7/3 volume of pure ethanol for an overnight precipitation at -20˚C. Dried pellets were resuspended in 10 mM Tris buffer and stored at -20˚C until further use. Quality assessment was performed for fragment size and concentration with Qubit (Invitrogen life Technologies) and TapeStation (Agilent Technologies) high sensitivity assays, according to the manufacturers' instructions. To confirm the specific presence of *Xci* in HERB_1937, we performed the *Xci*-exclusive Xac-qPCR diagnostic assay developed by Robène *et al.* on 3 replicates of 5 μL water-diluted (10 fold) DNA extract, our negative control and following the recommended amplification conditions [40].

## Library preparation & sequencing

Library preparation and sequencing were outsourced (https://www.fasteris.com/dna/). Briefly, DNA was converted into a double-stranded library using a custom TruSeq DNA Nano Illumina protocol omitting the fragmentation step and using a modified bead ratio to keep small fragments. Sequencing of both HERB_1937 and the negative control sample was performed in a paired-end 2×150 cycles configuration on a single lane of the NextSeq flow cell.

## Initial read trimming and merging

BBDuk from BBMap 37.92 [84] was first run with an entropy of 0.6 to remove artefactual homopolymer sequences. Illumina adaptors were trimmed out using the Illuminaclip option in Trimmomatic 0.36 [85]. Such roughly trimmed-reads were processed using the post-mortem DNA damage pipeline detailed below. Additional quality-trimming was performed with Trimmomatic 0.36 based on base-quality (LEADING:15; TRAILING:15; SLIDINGWINDOW:5:15) and read length (MINLEN:30). Paired reads were then merged using Adapter-Removal 2.2.2 [86] using default parameters before running both the metagenomic and the phylogenetic pipelines detailed below and in Fig 2.

## Negative control sequences analysis

Reads generated from the negative control sample were sequentially mapped to reference sequence genomes of *Coffea arabica* (GCA_003713225.1), *Citrus sinensis* (AJPS00000000.1)

and *Xci* (strain IAPAR 306, chromosome NC_003919.1, plasmids pXAC33 NC_003921.3 and pXAC64 NC_003922.1) using BWA-aln 0.7.15 (default options and seed disabled).

## Metagenomic pipeline

The metagenomic composition of historical HERB_1937 sample was assessed following a two-step procedure. First, reads were sequentially mapped to reference sequence genomes of human (GCF_000001405.39), *Citrus sinensis* (AJPS00000000.1) and *Xci* (strain IAPAR 306, chromosome NC_003919.1, plasmids pXAC33 NC_003921.3 and pXAC64 NC_003922.1) using the "very-sensitive" option (seed of–L 20) of Bowtie 2 [87]. In a second step, BLAST analysis was performed on 1,000,000 randomly chosen unmapped reads against the nucleotide database using the blastn command of NCBI BLAST 2.2.31 [88]. Only top hits with an e-value below 0.001 were saved. The proportion of each taxon in the sample was scaled over the total number of reads.

## Ancient DNA damage assessment pipeline

Post-mortem DNA damage measured by DNA fragment length distribution, purine frequencies before DNA breakpoints and 5' C to T or 3' G to A misincorporation patterns were assessed with mapDamage2 [89] for both the historical specimen and three modern strains (strains LJ225-01, LK144-08 & LM053-06 isolated in 2012, 2013 and 2015, respectively—see S1 Table). Alignments were generated using BWA-aln 0.7.15 (default options and seed disabled) [90] as short-read aligner for the historical specimen and Bowtie 2 (options—non-deterministic—very-sensitive) [87] for the modern strains using IAPAR 306 *Xci* reference genome (plasmids pXAC33, pXAC44 and chromosome). PCR duplicates were removed using picard-tools 2.7.0 MarkDuplicates [91]. An independent damage assessment was performed using *Methylobacterium* reference sequence (*Methylobacterium organophilum* strain DSM 760 QEKZ01000001.1) with BWA-aln (same options as above). Statistical analyses were performed using GraphPad Prism version 6.00 for macOS, GraphPad Software, San Diego, California USA (www.graphpad.com) [92].

## Historical genome reconstruction & characterization

Sequencing depths were computed using BEDTools genomecov 2.24.0 [93], and graphically represented with CIRCOS 0.69.9 [94]. BAM files were extremity-trimmed for 5 bp at each end with BamUtil 1.0.14 [95]. SNPs were called with GATK UnifiedGenotyper [96]. SNPs that met at least one of the following conditions: depth<average depth + 1sd [X = 9], allelic frequency<0.9, distance from another SNP<20 bp were considered as dubious and filtered out. Consensus historical sequences were then reconstructed by introducing the remaining high-quality SNPs in the reference genome and replacing both filtered-out variants and non-covered sites (depth = 0) by an N. Non-covered regions were identified with BEDTools 2.24.0 [93].

## Gene content analysis

The presence (or absence) of a CDS was assumed when its sequence coverage was found to be above (or below) a 75% threshold. Their repeated nature, as well as their hypothetical functions (as predicted for strain IAPAR 306, chromosome NC_003919.1, plasmids pXAC33 NC_003921.3 and pXAC64 NC_003922.1) were assessed using the annotated reference sequences within the genome browser and synteny tool of the MicroScope platform [97] based on a small set of public strains from the SWIO and the rest of the world (C40, LH201, LB100-

1, JJ10-1, FDC217, LG115, LG97, LB302 [33]), and a few additional representatives of the genus *Xanthomonas* (*X. citri* pv. *bilvae* strain NCPPB 3213, *X. euvesicatoria* 85–10, *X. campestris* pv. *campestris* 8004, *X. perforans* 91–118).

To investigate the presence of virulence factors in HERB_1937_Xci, we used a list of 82 Type III effectors (see list in S3 Table) found in *Xanthomonas* [45,46]. The reference sequences used to assess homology were the IAPAR 306 CDS when available or other *Xanthomonas* CDS for genes not present in *Xci*. We assessed coverage for the 57 effectors found in *Xci* from the reconstructed historical genome. For the 25 effectors from other *Xanthomonas*, reads were realigned on reference sequences with BWA-aln as described above. Coverage data was recovered from BAM files with BAMStats 1.25 tool [98].

In a second step, we aimed to specifically retrieve reads that initially did not map to *tale* genes. We performed a BWA-aln alignment (options as above) on the sequences coding for the conserved N- and C-terminal domains of the *pthA4* CDS from strain IAPAR 306 (S1 Fig). For reads corresponding to the central repeat domain, we constructed a chimera sequence of three repeats (containing Ns at the variable nucleotide positions in RVD) as a reference for the mapping.

## Phylogeny pipeline & tree-calibration

An alignment of HERB_1937_Xci and 116 modern genomes (date range: 1978–2015) from the SWIO islands was constructed for phylogenetic analyses, with the modern *Xci* strain LG117 from Bangladesh used as outgroup (CDAX01000000) (S1 Table). Variants from modern strains were independently called and filtered using the same parameters as for HERB_1937_Xci (except for the threshold on depth that was modified to a value of 15). Regions acquired via horizontal gene transfers were identified with ClonalFrameML [51] and removed to account for the effect of recombination on phylogenetic reconstruction and avoid incongruent trees. A Maximum Likelihood tree was constructed using RAxML 8.2.4 [52] using a rapid Bootstrap analysis, a General Time-Reversible model of evolution [99] following a Γ distribution with four rate categories (GTRGAMMA) and 1,000 alternative runs.

The existence of a temporal signal was investigated by two different tests. First, a linear regression test between sample age and root-to-tip distances (computed from the ML tree) was done using the distRoot function from the "adephylo" R package [100]. Temporal signal was considered present if a significant positive correlation was observed. Secondly, we performed a date-randomization test [101] with 20 independent date-randomized datasets using R package "TipDatingBeast" [102]. Temporal signal was considered present when there was no overlap between the inferred root height 95% Highest Posterior Density (95% HPD) of the initial dataset and that of 20 date-randomized datasets. Tip-dating calibration Bayesian inferences were performed with BEAST 1.8.4 [53]. For this, leaf heights were constrained to be proportional to sample ages. Flat priors (*i.e.*, uniform distributions) for the substitution rate ($10^{-12}$ to $10^{-2}$ substitutions/site/year), as well as for the age of any internal node in the tree, were applied. We also considered a GTR substitution model with a Γ distribution and invariant sites (GTR+G +I), an uncorrelated relaxed log-normal clock to account for variations between lineages, and a tree prior for demography of exponential growth as best-fit parameters described in Richard *et al.* [36]. The Bayesian topology was conjointly estimated with all other parameters during the Markov Chain Monte-Carlo and no prior information from the ML tree was incorporated in BEAST. Three independent chains were run for 25 million steps and sampled every 2,500 steps with a burn-in of 2,500 steps. Convergence to the stationary distribution and sufficient sampling and mixing were checked by inspection of posterior samples (effective sample size >200) in Tracer 1.7.1 [103]. Parameter estimation was based on the samples combined from the different chains. The best-supported tree was estimated from the combined samples by

using the maximum clade credibility method implemented in TreeAnnotator [53]. In order to assess the effect of including our historical sample in the tree calibration, we computed the same inferences on a dataset excluding HERB_1937_Xci. Wilcoxon rank sum test with continuity correction and a Bartlett test of homogeneity of variances were performed on the posterior estimates of the tree root age, to respectively compare the mean and variance of this parameter from both datasets. Finally, phylogenetic diversity (PD) [104], calculated as the sum of branch lengths of the minimum spanning path between strains of the region (island, or group of islands in the case of the Comoros) was calculated on patristic distances from the reconstructed phylogeny using the distRoot function implemented in "adephylo" R package [100]. To account for heterogeneity in region samplings, they were down-sampled to the smallest sampling (Mayotte = 10) and PD by region averaged over 1,000 iterations. PD comparison was done using a Wilcoxon rank sum test with continuity correction.

## Supporting information

**S1 Fig. Reads depth of a Transcription Activator-Like Effector (TALE) gene of HERB_1937.**
(PDF)

**S2 Fig. Maximum Likelihood (ML) phylogenetic tree of *Xci* genomes.**
(PDF)

**S3 Fig. Date-randomization test results.**
(PDF)

**S4 Fig. Effect of integrating HERB_1937_Xci on substitution rate estimates in BEAST.**
(PDF)

**S1 Table. Published modern genomes included in the phylogenetic analyzes.**
(PDF)

**S2 Table. List of *Xci* reference strain IAPAR 306 coding sequences (CDS) covered on less than 75% of their length by HERB_1937_Xci reads and hence designed as non-covered.**
(PDF)

**S3 Table. List and coverage of 82 *Xanthomonas* virulence factors CDS (pthA4 not included) used in this study.**
(PDF)

**S4 Table. List and frequency of nucleotide patterns coding for RVD found in HERB_1937_Xci reads.**
(PDF)

**S5 Table. Description of the 14 SNPs found in coding regions between HERB_1937_Xci and modern strains of the SWIO clade.**
(PDF)

## Acknowledgments

We are grateful to F. Chiroleu, A. Doizy, A. Duvermy, P. Lefeuvre, F. Balloux, V. Llaurens, R. Debruyne, A. Pérez-Quintero & I. Robène for valuable comments and discussions. Computational work was performed on the CIRAD HPC data center of the South Green bioinformatics platform (http://www.southgreen.fr/)

## Author Contributions

**Conceptualization:** Paola E. Campos, Philippe Roumagnac, Boris Szurek, Nathalie Becker, Lionel Gagnevin, Adrien Rieux.

**Data curation:** Paola E. Campos, Adrien Rieux.

**Formal analysis:** Paola E. Campos, Nathalie Becker, Lionel Gagnevin, Adrien Rieux.

**Funding acquisition:** Philippe Roumagnac, Boris Szurek, Nathalie Becker, Lionel Gagnevin, Adrien Rieux.

**Investigation:** Paola E. Campos, Clara Groot Crego, Karine Boyer, Damien Richard, Olivier Pruvost, Nathalie Becker, Lionel Gagnevin, Adrien Rieux.

**Methodology:** Paola E. Campos, Nathalie Becker, Lionel Gagnevin, Adrien Rieux.

**Resources:** Myriam Gaudeul, Claudia Baider.

**Supervision:** Nathalie Becker, Lionel Gagnevin, Adrien Rieux.

**Visualization:** Paola E. Campos, Adrien Rieux.

**Writing – original draft:** Paola E. Campos, Clara Groot Crego, Nathalie Becker, Lionel Gagnevin, Adrien Rieux.

**Writing – review & editing:** Paola E. Campos, Clara Groot Crego, Claudia Baider, Damien Richard, Olivier Pruvost, Philippe Roumagnac, Boris Szurek, Nathalie Becker, Lionel Gagnevin, Adrien Rieux.

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
