## [Decision Letter · Decision Letter 0]

9 Feb 2021

Dear Dr Rieux,

Thank you very much for submitting your manuscript "First historical genome of a crop bacterial pathogen from herbarium specimen: insights into citrus canker emergence" for consideration at PLOS Pathogens.

First off, I'd like to apologize for length of time that has passed since this paper was submitted. Difficulty in recruiting reviewers and receiving reviews, compounded by the holiday break, resulted in the delay.

As with all papers reviewed by the journal, your manuscript was reviewed by members of the editorial board and by several independent reviewers. In light of the reviews (below this email), we would like to invite the resubmission of a significantly-revised version that takes into account the reviewers' comments.

We cannot make any decision about publication until we have seen the revised manuscript and your response to the reviewers' comments. Your revised manuscript is also likely to be sent to reviewers for further evaluation.

Sincerely,

David Mackey

Associate Editor

PLOS Pathogens

Wenbo Ma

Section Editor

PLOS Pathogens

Kasturi Haldar

Editor-in-Chief

PLOS Pathogens

orcid.org/0000-0001-5065-158X

Michael Malim

Editor-in-Chief

PLOS Pathogens

orcid.org/0000-0002-7699-2064

Reviewer's Responses to Questions

**Part I - Summary**

Reviewer #1: The manuscript entitled “First historical genome of a crop bacterial pathogen from herbarium specimen: insights into citrus canker emergence” by P.E. Campos and co-authors reports for the first time the description of the genome of a plant bacterial pathogen from a herbarium specimen. The authors targeted a 1937 specimen of Citrus sp. from Mauritius island (located in South West Indian Ocean, SWIO) showing typical symptoms of Asiatic citrus canker, an economically important disease caused by Xanthomonas citri pv. citri (Xci). They first confirmed the historical nature of the reconstructed genome HERB_1937 Xci by characterizing the DNA degradation patterns specific to ancient DNA. The authors further described the associated metagenome and again excluded recent contaminations by looking at the aDNA patterns of the most predominant taxon of the associated bacterial community. They investigated the HERB_1937_Xci gene content both on the chromosome and the two plasmids focusing on virulence factors and revealed a full equipment for virulence and detected three new allele sequence in TALE genes that are important for virulence. Then the authors compared the HERB_1937_Xci genome with a large set of modern genomes, especially including many genomes originating from the SWIO region. They performed a phylogenetic reconstruction which confidently placed HERB_1937_Xci at the root of the modern SWIO strains lineages making him a very likely member of the founder population. Having ensured the presence of a temporal signal, the authors finally estimated evolutionary parameters using a tip-calibration inference/ Bayesian tip-calibration inferences and refined the estimation of the age of the ancestor of modern SWIO strains proposing a scenario for the emergence of ACC in the SWIO region.

The authors have taken the opportunity of an immense resource of biodiversity that lies in the herbaria to increase our present knowledge on genetic diversity of a bacterial plant quarantine pathogen that still threatens large areas of citrus production. They reconstructed the introduction routes of this invasive bacterial species in SWIO. The evolutionary history that can be deduced provides information about the origin and history of the ACC emergence and the construction of the genetic diversity of the SWIO Xci populations.

They have produced original and new results that underline the interest of the methodological approach. From what I can judge, this methodological approach and the analyses implemented are appropriate and relevant to reach the conclusions presented. The manuscript is well written and the results are clearly presented.

Reviewer #2: Strength

First study to sequence the whole bacterial genome of an important plant pathogen in a historic herbarium sample

Data analysis was thorough and paper is well- written

Weakness

Could have sequenced more specimens to address migration and root tree and examine evolution for genome over time.

Did not justify why they chose this particular sample.

Reviewer #3: This manuscript makes another contribution to a relatively underexplored area in ancient DNA – herbarium specimens from historical collections that show indications of infection. I was asked to comment specifically on the measures of authenticity regarding ancient DNA acquisition and analysis, so my comments largely relate to that. These comments aside, the paper is very clearly written and the analyses are extensive. I include a few comments below that might improve quality of the analysis.

**Part II – Major Issues: Key Experiments Required for Acceptance**

Reviewer #1: No major issues to be discussed here.

Reviewer #2: Need to sequence more historic genomes to justify some of their conclusions. That would make the paper stronger. They included a lot of previously sequenced genome data from modern lineages. For this work, I would ask them to qualify their conclusions and not do additional experiments

Reviewer #3: First, it is typical in ancient DNA studies to eliminate all reads 30bp and shorter, after trimming (if this is done, which it is here). It seems the authors have not done so, despite their caution against it (line 172).

The damage plot they present in Figure 4b is also confusing. The use of two horizontal axes, one on the top and one on the bottom, is confusing, as they are clearly independent, though at first glance they look as though they are describing the damage profiles of all reads 25bp in length. A more intuitive way to show their damage profiles would be to show the 5’and 3’ plots side by side. They could show only the first 25bp of the read in either direction, but this removes the ambiguity of the relationship between the positions in the 5’ and 3’ ends as shown in the current plot.

Line specific comments:

Line 61: Change “fossil remains” to “preserved tissues” or something similar, as fossils are rock with no preserved biological tissues

Line 105 – a sentence or two describing the preservation conditions of the sample would be helpful

Line 109 – it’s not clear if the reads were merged, or simply treated as regular mate pair data. Reads should be merged, as treatment as modern paired end data would lead to incorrect base calling due to high background DNA. Treatment of each read independently will result in SNP calls in duplicate reads (via a call in both the forward and reverse read) and will inflate SNP coverage, which could be an issue with high non-target background. I suggest they merge their reads and reprocess the data.

Line 116 – Greater clarity is needed here to explore how these percentages were generated. I assume via BLAST? What seems to be missing is a description of the percent Xci DNA in bulk DNA content based on mapping to a reference. This is a key value in aDNA studies.

Line 127 – Remove or edit the passage “as compared to the standard read length of 150 nt obtained for modern Xci samples”, since read length is dependent on the sequencing platform and kit, and that is not clear based on this description.

Line 144 – 148: I’m not sure why the authors opted to make this comparison. A far more informative metric would be to compare the Xci DNA damage profile to that of the host (perhaps to a plastid or a chromosome). That is typically done as a means of authenticity in ancient pathogen investigations.

Table 1: Column 2 should be % of reads mapping to the reference based on total DNA content (% endogenous DNA), before duplicate removal. Currently this represents the proportion of mapped reads that assigns to the genetic component, which isn’t as informative. Column 3 is called “Mean depth”, though I suppose they mean “Mean coverage”, which is a clearer term. Last column, which shows the % reads with damaged terminal bases show values between 1.62 and 1.88, though 2.64% is reported in the main text (line 135). Why the discrepancy?

Lines 168 – 169: Reporting % of the reference covered at 1X is not terribly informative, especially if base calling required a min of 15-fold coverage (as is described in the methods). Also is it not clear if the map quality was 0 or higher. If it was 0, the authors are reporting reads that will map to multiple locations in the genome, which could inflate their coverage, especially if these positions are later removed for SNP calling.

Line 273 – “obtained from such material”… greater specificity is needed here to describe what “such material” is.

Line 277 – “contamination-free environment”: no environment is contamination-free.

Lines 291 – 293: There is no need to explore the reasons for 60% of the reads not mapping to a reference. This is a nearly ubiquitous phenomenon in ancient DNA work, and probably also in modern metagenomics studies.

Line 295 – change “studies” to “study”

Line 297 – were the preservation conditions for P. infestans similar?

Line 298 – “in the reads mapping to either of two plasmids” is a better way to say this

Lines 331 – 334: If a conclusion is being drawn from tree topology, the ML tree with bootstraps should ideally be included in the main text. Currently the authors include only the Bayesian tree, which was constructed with the ML tree as a prior (if I have understood the methods properly)

Line 361 – typo “deshydrogenase”

Lines 392 – 403: There is some odd grammar in this concluding paragraph, and the English should be adjusted for better clarity

**Part III – Minor Issues: Editorial and Data Presentation Modifications**

Reviewer #1: Minor comments are indicated below

The first three points are free comments and do not necessarily need to be followed up on.

- Line 45. And to possibly close the loop, loss of biodiversity can impact disease risk (Keesing, 2006, Ecol. Lett. 9:485; Rohr et al., 2020, Nature Ecol. & Evol. 4:24).

- May be you could refer to the pioneering work of Ristaino et al (2001, Nature 411:695).

- Line 91. You could add a sentence highlighting further the contribution of herbarium specimens in deciphering the original outbreak of ACC in Florida at the beginning of the twentieth century (Li et al., 2007).

- Line 105 (and line 416). DNA was extracted from a pool of five lesions which could represent independent infection events. Is there any possibility of strains mixture? Or did you observe any sequence variation within HERB_1937_Xci that could be attributed to the presence of different strains?

- Line 115. You talk about identified sequences at the species level but the taxonomic level of the plant host is at the genus level. Even if this was not your objective, did you get any clues from the reads about the species identification of your citrus specimen?

- Line 199. Change to S3 Table for homogenization.

- Line 261. Check with Fig 6b where R2 = 0.043 and p-value = 0.0105.

- Line 312. Did you estimate any genetic diversity indices to support this assertion?

- Line 336. Can a privileged ancient history between these two islands explain older migration events and shed light on the presence of older Xci common ancestors?

Bibliography

- Line 645. Change to canker-like.

- Line 743. Reference Bolger et al. to be completed.

- Line 776. Prefer: Jombart T., Balloux F., Dray S., adephylo: new tools for investigating the phylogenetic signal in biological traits, Bioinformatics, Volume 26, Issue 15, 1 August 2010, Pages 1907–1909, https://doi.org/10.1093/bioinformatics/btq292

I could not consult the data associated with HERB_1937 deposited to the SRA and on GenBank database which were not yet available.

Reviewer #2: Page Line Comments

Abstract 1 Change to “ancient genomics has been used”

3 43 Add a . after (3) and start new sentence with “More”

3 46 Change to “infectious crop” diseases

3 54 Rework this sentence as there have been studies with herbarium collections that sampled over more than 4 decades. See paper by Ristaino et al “Ristaino, J. B. 2020. The importance of mycological and plant herbaria in tracking a plant killer. Front. Ecol. Evol. 7:521. doi: 10.3389/fevo.2019.00521” for a review of the studies with plant mycological collections to track plant disease epidemic”.

3 58 Delete “to” and change to “systemic detection of evolutionary changes or reconstruction of …”

4 63 Add more citations here. Yoshida was not the first to use herbarium specimens to identify the famine-era lineage of P. infestans. Cite “May, K. J. and Ristaino, J. B. 2004. Identity of the Mitochondrial DNA Haplotype(s) of Phytophthora infestans in Historical Specimens from the Irish Potato Famine. Mycol. Res. 108:171-179.”

4 81 Add more citations here “Saville, A. , Martin, M, Gilbert, T. and Ristaino, J. 2016. Historic late blight outbreaks caused by a widespread dominant lineage of Phytophthora infestans (Mont.) de Bary Plos One 11: e0168381 https://doi.org/10.1371/journal.pone.0168381” and others mentioned previously. I’d like to think our work has helped resolved the debate about the identity and source of the 19th century outbreak.

5 84 Cite Li et al. (26) here as Hartungs work did use herbarum specimens to study citrus canker and I believe did the first genetic work with a historic strain from 1911. Although they did not use full genomes sequences they did to multilocus genotyping so say a bit more here and compare your results to previous work in the discussion.

5 104 Indicate the name of the herbarium that the sample was retrieved from and define SWIO.

6 116 There was a greater percentage of reads from some other bacterial species than X. citir (1.2% vis Methylbacterium 1.57%) Methylobacterium has been identified as a contaminant during DNA extraction and can lead to its erroneous appearance in microbiota or metagenomic datasets. Did you run a preliminary PCR on the samples to confirm they were actually infected with X citri before the llumina sequencing using species specific primers. Rep or Box PCR, etc It would be good to conform or look for sequences specific to the pathogen to confirm that the samples were infected by X citri.

6 122 Could reduce some of this section about DNA damage as its clear you are using herbarium material and the read will likely be short.

9 175 Did you sequence more than one herbarium sample? Why was this particular sample of interest. Is the oldest existing specimen? It would have been better to sequence more samples from the various islands and at different times in the past to present day.

10 199 The data on the T3SS is interesting. Were the absent genes viewed necessary for virulence in modern lineages?

10 215 This mapping method that uncovered new reads in the TALE region is interesting

11 227 Say more about the number and location of the SNPS. What do you mean by “dubious SNPS”?

12 268 I can’t help but think that sequencing more samples would also have improved the root date of the tree. What do you know about movement of citrus during this time period? Are there historical records on the introduction of the pathogen that might indicate times of introduction?

13 274 Add more citations here as stated above on page 3 line 54.

13 284 Sequencing more samples would have given you an indication of whether this was contamination or infection by other bacteria.

14 320 Are the French botanists samples available in the Paris herbarium? These would be interesting to see if you can push back the introduction date.

15 331 You really need more samples to determine if one or more introduction occurred or even it the same pathovar was introduced. This is speculation so reword this.

15 333 You could do migration analysis on the data to see if directional gene flow has occurred among the island nations.

16 363 How would this influence pathogenicity – more or less pathogenic? Is the historic lineage potentially more virulent then modern day lineages or not?

17 374 I encourage you to continue more sequencing of historic genomes of this important pathogen so you can try de novo genome assembly and do further migration and phylogenetic analysis . Search for specimens in unlikely places. They may be in museum collections from explorers to the region including the British.

Reviewer #3: (No Response)

PLOS authors have the option to publish the peer review history of their article (what does this mean?). If published, this will include your full peer review and any attached files.

Reviewer #1: No

Reviewer #2: No

Reviewer #3: No
---

## [Decision Letter · Decision Letter 1]

13 May 2021

Dear Dr Rieux,

Thank you very much for submitting your manuscript "First historical genome of a crop bacterial pathogen from herbarium specimen: insights into citrus canker emergence" for consideration at PLOS Pathogens. This revised version of your manuscript was reviewed by one of the previous independent reviewers. They make several suggestions to further improve the manuscript. Based on the reviews, we are likely to accept this manuscript for publication, providing that you modify the manuscript according to the review recommendations. Please take special care to address the concern regarding in-lab contamination and the need to report on mapping reads in the negative control.

Sincerely,

David Mackey

Associate Editor

PLOS Pathogens

Wenbo Ma

Section Editor

PLOS Pathogens

Kasturi Haldar

Editor-in-Chief

PLOS Pathogens

orcid.org/0000-0001-5065-158X

Michael Malim

Editor-in-Chief

PLOS Pathogens

orcid.org/0000-0002-7699-2064

Reviewer Comments (if any, and for reference):

Reviewer's Responses to Questions

**Part I - Summary**

Reviewer #3: (No Response)

**Part II – Major Issues: Key Experiments Required for Acceptance**

Reviewer #3: (No Response)

**Part III – Minor Issues: Editorial and Data Presentation Modifications**

Reviewer #3: I appreciate that the authors took the time to reprocess their data as merged, as this did have an effect on their coverage estimates. I’m slightly curious as to why the merged data resulted in a greater number of genomic regions with no coverage, as I would have thought that merging would result in a reduction of reads spanning only those positions that were covered via paired end analysis. Was the map quality metric altered between these two analyses? Certainly increasing the map quality could lead to this phenomenon. Typically map quality values of 30 or even 37 are used, not just removal of reads with map quality 0. Regarding coverage thresholds for SNP calling, the authors have now used average depth as a criterion for inclusion in the analysis, citing a study (ref 1 of their response) as a demonstration of this approach. While this approach is acceptable in their case, it should not be considered a global criterion for ancient DNA work, where coverage is often highly variable and can be very low. It is customary to set a threshold independent of coverage such as 3X or better yet 5X for SNP calling with ancient data.

I thank the authors for including their percent Xci DNA in bulk content (0.74%). I have not seen them disclose their sequencing depth anywhere in the manuscript, and that should be mentioned ideally in the main text. Their genomic reconstruction was performed from 1.6 million mapping reads, so by extension I assume their sequencing depth was ca. 220 million reads. I’m also curious about why the BLAST analysis assigned 1.2% of bulk DNA to Xci. I would have thought BLASTn would have returned fewer reads assigned to Xci compared to mapping, since the mapping process is based on read similarity only. Was a filtering step applied before the mapping? A comment about this difference could be useful.

A description of the mapping process and statistics of the Xci mapping should be presented in the main text before the damage assessment, which is calculated based on the mapped reads. They should also be clear on line 146 that their patterns of degradation apply to reads mapping to the Xci reference genome as opposed to simply “degradation in HERB_1937” as is it currently stated. Why are the damage profiles now shown in both the main manuscript (for the chromosome) and the SI (for the chromosome and two plasmids)? Why not just show the current SI image in the main text? I also caution the authors against making assumptions that damage alone will “itself prove the authenticity” (wording from response letter) of their data. If they were to evaluate the human reads and observe a similar damage pattern, the data would have to be explained by contamination. Authenticity is conferred through damage profile along with context of the finding. Branch shortening in the phylogeny can be a useful metric for authentication of ancient data, and this is shown in their supplementary ML tree. The authors may wish to mention that as well at some point in the MS to bolster their claim of authenticity, along with highlighting its ancestral phylogenetic position (currently on line 329 outside the context of authenticity). They entertain the possibility on lines 303 – 304 of their data coming from in-lab contamination, but I don’t think this statement is needed in light of the accompanying authenticity data they present. Reporting on the mapping reads in their negative controls is currently missing in the main manuscript and absolutely must be included: This should also help to rule out in-lab contamination.

The resolution of the main text images I received is of poor quality, and it looks like the samples have 20% damage as opposed to 2% damage, which really shocked and confused me at first. Please make sure that high resolution figures accompany the manuscript in print.

In their table 1 (main MS), I assume “most extreme position” refers to the terminal position? I’ve not come across that terminology and I find it unintuitive. Also, it is not common for % damaged bases to be presented as an average over the terminal five nucleotides, as the authors have done here. % damage at the terminal 5’ and 3’ ends is sufficient and is more commonly reported.

Is it worth mention that the reads mapping to Xci are shorter than average for bulk DNA content and those mapping to Methylobacterium are longer than average? Do the authors know of a biological reason for this difference? GC content perhaps? Cellular structure?

PLOS authors have the option to publish the peer review history of their article (what does this mean?). If published, this will include your full peer review and any attached files.

Reviewer #3: No

Figure Files:

Data Requirements:

Reproducibility:

References:

---

## [Editor Report · Decision Letter 2]

14 Jun 2021

Dear Dr Rieux,

We are pleased to inform you that your manuscript 'First historical genome of a crop bacterial pathogen from herbarium specimen: insights into citrus canker emergence' has been provisionally accepted for publication in PLOS Pathogens. The careful and thorough responses to review comments was much appreciated.

Best regards,

David Mackey

Associate Editor

PLOS Pathogens

Wenbo Ma

Section Editor

PLOS Pathogens

Kasturi Haldar

Editor-in-Chief

PLOS Pathogens

orcid.org/0000-0001-5065-158X

Michael Malim

Editor-in-Chief

PLOS Pathogens

orcid.org/0000-0002-7699-2064
---

## [Editor Report · Acceptance letter]

6 Jul 2021

Dear Dr Rieux,

We are delighted to inform you that your manuscript, "First historical genome of a crop bacterial pathogen from herbarium specimen: insights into citrus canker emergence," has been formally accepted for publication in PLOS Pathogens.

Best regards,

Kasturi Haldar

Editor-in-Chief

PLOS Pathogens

orcid.org/0000-0001-5065-158X

Michael Malim

Editor-in-Chief

PLOS Pathogens

orcid.org/0000-0002-7699-2064